# Prediction of Casing Collapse Strength Based on Bayesian Neural Network

Dongfeng Li [1,2], Heng Fan [3,*], Rui Wang [2], Shangyu Yang [2], Yating Zhao [3] and Xiangzhen Yan [1]

[1] College of Pipeline and Civil Engineering, China University of Petroleum (East China), Qingdao 266580, China; viviwing_lee@sina.com.cn (D.L.); yanxz163@163.com (X.Y.)
[2] CNPC Tubular Goods Research Institute, Xi'an 710077, China; wangrui018@cnpc.com.cn (R.W.); yangshangyu@cnpc.com.cn (S.Y.)
[3] School of Electronic Engineering, Xi'an Shiyou University, Xi'an 710077, China; zhao_yating0818@126.com
* Correspondence: fan_h@xsyu.edu.cn

**Abstract:** With the application of complex fracturing and other complex technologies, external extrusion has become the main cause of casing damage, which makes non-API high-extrusion-resistant casing continuously used in unconventional oil and gas resources exploitation. Due to the strong sensitivity of string ovality, uneven wall thickness, residual stress, and other factors to high anti-collapse casing, the API formula has a big error in predicting the anti-collapse strength of high anti-collapse casing. Therefore, Bayesian regularization artificial neural network (BRANN) is used to predict the external collapse strength of high anti-collapse casing. By collecting full-scale physical data, including initial defect data, geometric size, mechanical parameters, etc., after data preprocessing, the casing collapse strength data set is established for model training and blind measurement. Under the classical three-layer neural network, the Bayesian regularization algorithm is used for training. Through empirical formula and trial and error method, it is determined that when the number of hidden neurons is 12, the model is the best prediction model for high collapse resistance casing. The prediction results of the blind test data imported by the model show that the coincidence rate of BRANN casing collapse strength prediction can reach 96.67%. Through error analysis with API formula prediction results and KT formula prediction results improved by least square fitting, the BRANN-based casing collapse strength prediction has higher accuracy and stability. Compared with the traditional prediction method, this model can be used to predict casing strength under more complicated working conditions, and it has a certain guiding significance.

**Keywords:** casing collapse strength; Bayesian regularization algorithm; artificial neural network

## 1. Introduction

As a key component in the development and production of oil and gas wells, the casing is not only subjected to high axial tensile or compressive loads, as well as internal and external pressure loads, but also the harsh service environment such as high-temperature environment at the bottom of the well and acidic corrosion. Once damage occurs, it will not only reduce oil and gas production but also seriously damage the reservoir and affect the normal exploration and production [1–3]. The economic loss of well damage or scrapping caused by casing damage in China's oil fields amounts to billions of dollars every year, and up to now, the casing damage problem is still a non-negligible part of the international oil industry. Due to the long-term complex service environment, the casing is subject to various uniform or non-uniform loads formed by the formation and downhole operations, and its full-scale performance will constantly change due to the transformation of the mechanical environment and downhole working conditions. Numerous studies on casing strength show that casing steel grade, diameter-to-thickness ratio, geometric defects (OD ellipticity and wall thickness unevenness), yield strength, and residual stresses are the main factors affecting casing collapse strength. In addition, external factors such as temperature,

downhole wear, and cement rings also affect the casing collapse strength [4]. In recent years, with the deepening of drilling depth, the casing collapse strength has become a key indicator in casing selection. The calculation of collapse strength used API and ISO standards cannot fully consider the relationship between the inherent defects of the tubular column and the non-uniform external load in complex downhole conditions, resulting in the calculated value deviating from the actual value. To investigate the change law of casing strength performance under the influence of multiple factors, scholars at home and abroad have revised the formula for calculating the casing resistance to collapse with experiments and finite element simulations. At present, the research on collapse strength is still being explored, seeking a more accurate formula for casing collapse strength prediction from a data-driven perspective.

Big data analytics, as a branch of data science, covers artificial intelligence, data mining, machine learning, and pattern recognition. Machine learning studies the ability of computers to learn based on data and is used to extract predictive models from data [5–9]. Machine learning is divided into two types. Supervised learning, which learns by classifying or labeling data, and supervised learning, which analyzes trained data to obtain a model capable of predicting new cases based on a vector of homogeneous features [10,11]. Artificial neural networks are one of the most widely used machine learning methods in the oil and gas industry, with applications in oilfield production, drilling, fluid processing, etc. They are a form of a mathematical structure inspired by biological neural networks for approximating functions that rely on large amounts of input data. Neural networks "learn" from samples and identify associations between input and output values from a selected sequence of data [12–15]. Since there is no specific expected value for the correlation between the data and the physical properties of each parameter in the model are independent, the collapse strength of the casing can be predicted by combining different process parameters. In this paper, the main correlation factors of the current casing collapse strength are combined with the relevant data obtained from the laboratory collapse experiments, and the data samples are formed after data pre-processing for training the artificial neural network to form the casing collapse strength prediction model. In terms of algorithm optimization, the Bayesian regularization algorithm is considered to improve the generalization ability of the model and further ensure the effectiveness of the model.

## 2. Prediction Model Scheme of Casing Collapse Strength Based on Bayesian Regularization Algorithm

*Model Construction Scheme*

Casing manufacturing process defects and complex underground service environment make casing become one of the weak links in the oil and gas industry. Combining with the relevant specifications of casing strength design, among many factors affecting casing strength performance, diameter-thickness ratio, ovality, wall thickness unevenness, yield strength, and residual stress are selected to carry out the prediction research on collapse strength. Figure 1 shows the flow chart of the scheme for predicting casing collapse strength with the help of an artificial neural network, which includes three parts: data acquisition, model development, and comparative analysis of prediction results.

In the data acquisition part of the scheme, the full-scale physical performance experiment will be used to acquire enough data on collapse strength and establish data sets. The model development will be realized from the structure of the neural network, the division of the data set, the optimization of the model, and the evaluation of the model. Finally, in order to ensure that the model can effectively predict the collapse strength, the reserved test data is imported for prediction, and the accuracy of the predicted values of the model is calculated and compared by combining the traditional regression fitting method and the collapse strength calculation formula in API 5C3 specification, so as to test the validity of the model.

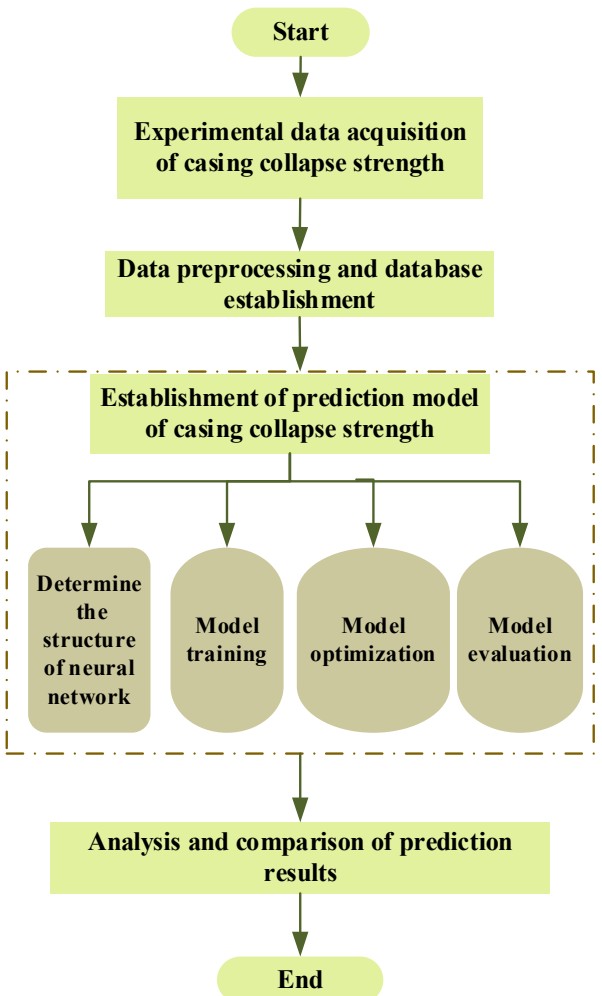

**Figure 1.** Flow chart of prediction scheme of casing collapse strength.

## 3. Bayesian Regularized Artificial Neural Networks

In artificial neural networks, neural information is stored in the form of weights and biases, and the magnitude of the weight value determines the impact of the corresponding information on the whole model. The classical machine learning approach divides a data set into 3 parts: training data subset, validation data subset, and test data subset [16,17]. Bayesian regularized neural networks refer to networks that use Bayesian regularization methods to train BP. Bayesian-Regularization (BR) refers to the process of improving the generalization ability of a neural network by modifying its performance function [10,18,19]. For function approximation, the most commonly used is the multilayer perception (MLP) algorithm with backpropagation. The MLP architecture based on the BP algorithm is given in Figure 2 and is the basis for developing the BRANN-based model in this study.

It can be seen that the architecture consists of 3 key components: the input layer, the implicit layer, and the output layer. The signal ($X_i$) in the input layer is first passed through a series of weights ($w_{xi},i$) and then passed into the implied layer through a commonly used activation function (e.g., logistic function or hyperbolic tangent function). Therefore, these processed signals are passed through using another series of weights ($w_{yi}$) and eventually summed to an output (YI) by a linear transfer function in the output layer. The mean squared error function ED is then iteratively calculated to determine the optimal weights and ultimately the appropriate architecture.

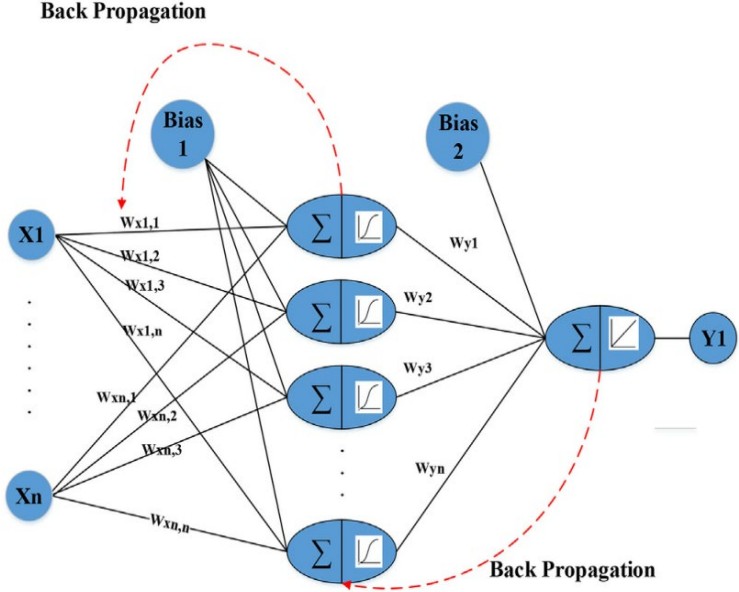

**Figure 2.** MLP architecture based on BP algorithm.

However, the traditional BP algorithm may encounter overfitting problems, i.e., small bias and large variance. As an alternative method, BRANN has better generalization capability. The objective function F (including the combination of the mean square error function ED and the weight decay function EW) is minimized and the optimal weights and objective function parameters are fitted in a probabilistic manner. The objective function of BRANN is:

$$F = \beta E_D + \alpha E_w \tag{1}$$

$$E_D = \frac{1}{N} \sum_i^N (y_i - t_i)^2 = \frac{1}{N} \sum_i^N e_i^2 \tag{2}$$

$$E_w = \frac{1}{2} \sum_i^m w_i^2 \tag{3}$$

where $\alpha$ and $\beta$ denote hyperparameters to control the distribution of other parameters. $w$ is the weight and $m$ is the number of weights. $D = (x_i, t_i)$ denotes the data of the training set with $i = 1, 2, \dots, N$, where $N$ is the total number of training sets (input-output pairs). $y_i$ denotes the $i$th output value corresponding to the $i$th training set (input-output pairs).

In BRANN, the initial weights are set randomly. With these initial weights, the density function of the weights can be updated according to Bayer's rule:

$$P(w|D, \alpha, \beta, M) = \frac{P(D|w, \beta, M) \cdot P(w|\alpha, M)}{P(D|\alpha, \beta, M)} \tag{4}$$

where $M$ is the particular neural network architecture used; $P(w|\alpha, M)$ is the prior density, which represents the knowledge of the weights before collecting the data; $P(D|w, \beta, M)$ is the likelihood function, which is the probability of the data occurring given a weight $w$; and $P(D|\alpha, \beta, M)$ is the normalization factor, which can be calculated by the following equation:

$$P(D|\alpha, \beta, M) = \int_{-\infty}^{+\infty} P(D|\alpha, \beta, M) P(w|\alpha, M) dw \tag{5}$$

If the noise of the training set data and weights is assumed to be Gaussian distributed, the probability density can be calculated by:

$$P(D|\alpha, \beta, M) = \frac{1}{Z_D(\beta)} exp(-\beta E_D) = (\pi/\beta)^{-N/2} exp(-\beta E_D) \tag{6}$$

$$P(w|\alpha, M) = \frac{1}{Z_w(\alpha)} exp(-\beta E_W) = (\pi/\alpha)^{-m/2} exp(-\beta E_W) \tag{7}$$

If these probability densities are substituted into Equation (4). The probability equation becomes:

$$P(w|D, \alpha, \beta, M) = \frac{\frac{1}{Z_W(\alpha)} \frac{1}{Z_D(\beta)} exp(-(\beta E_D + \alpha E_W))}{P(D|\alpha, \beta, M)}$$
$$= \frac{1}{Z_F(\alpha, \beta)} exp(-F(w)) \tag{8}$$

In BRANN, determining the optimal weights means maximizing the posterior probability $P(w|D, \alpha, \beta, M)$, in this case, the objective function $F$ of the minimization regularization. Combined post-test density:

$$P(\alpha, \beta/D, M) = \frac{P(D/\alpha, \beta, M) \cdot P(\alpha, \beta/M)}{P(D/M)} \tag{9}$$

The maximized joint posterior density can be determined by maximizing the likelihood function $P(D/\alpha, \beta, M)$, which is calculated as follows:

$$P(D/\alpha, \beta, M) = \frac{P(D/w, \beta, M) \cdot P(w/\alpha, M)}{P(w/D, \alpha, \beta, M)} = \frac{Z_F(\alpha, \beta)}{(\pi/\beta)^{\frac{n}{2}} (\pi/\alpha)^{\frac{m}{2}}} \tag{10}$$

where $n$ is the number of observations (input target simulation pairs) and m is the total number of network parameters. In addition, the parameter $Z_F(\alpha, \beta)$ depends on the Hessian of the objective function (prevent and Hagan, 1997), which is calculated as follows:

$$Z_F(\alpha, \beta) \propto \frac{e^{-F(wmax)}}{\sqrt{|Hmax|}} \tag{11}$$

where the subscript "*max*" indicates the maximum posterior probability. The Hessian matrix ($H$) is calculated from the Jacobian ($J$) as follows:

$$H = J^T J \tag{12}$$

where the Jacobi matrix contains the first-order derivatives of the network errors concerning the network parameters.

## 4. Experimental Data Acquisition

Full-scale collapse performance is a key parameter in ensuring the quality and safe use of the casing [20–22]. The full-scale collapse tests were carried out by the requirements of API RP 5C5 and API TR 5C3. The standard specifies a minimum length of 8 times the nominal outside diameter (D) for tubes with a nominal outside diameter (D) less than or equal to 9–5/8in and 7 times the nominal outside diameter (D) for tubes with nominal outside diameter (D) more than 9–5/8in [23]. The collapse test is carried out utilizing a composite collapse test system, which requires full-scale specimen lengths and no radial or axial loads. The composite collapse test system ensures that the specimen is slowly depressurized after the collapse has occurred and that the error does not exceed 1% of the collapse test pressure.

The collapse test specimens were geometrically measured before the test and the measurement locations are shown in Figure 3. Five sections were measured for each specimen and 8 points were measured for each section. The results of the geometric

measurements for specimen #1 are shown in Table 1. The average outer diameter, average wall thickness, ellipticity, and wall thickness unevenness values were calculated.

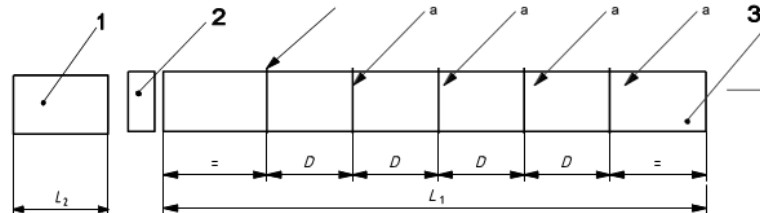

**Figure 3.** Measured specimen before collapse test. Note: 1 residual stress test specimen; 2 tensile specimens; 3 collapseed specimen. $L_1$ minimum length of the collapsed specimen. $L_2$ minimum length of residual stress specimen. Average outside diameter, average wall thickness, and ellipticity are measured at five equally spaced locations and the wall thickness unevenness is calculated from the wall thickness measurements.

**Table 1.** Specimen geometry inspection results (mm).

| Measurement section | | | | Average outside diameter | Ellipticity (1) |
|---|---|---|---|---|---|
| M1-N1 | G1-H1 | O1-P1 | E1-F1 | | |
| 141.58 | 141.18 | 141.07 | 141.17 | 141.25 | 0.36 |

| M1 | N1 | G1 | H1 | O1 | P1 | E1 | F1 | Average wall thickness | Wall thickness unevenness (2) |
|---|---|---|---|---|---|---|---|---|---|
| 13.09 | 13.43 | 12.77 | 13.46 | 12.96 | 12.84 | 13.16 | 13.15 | 13.11 | 5.26 |

| Measurement section | | | | Average outside diameter | Ellipticity |
|---|---|---|---|---|---|
| M2-N2 | G2-H2 | O2-P2 | E2-F2 | | |
| 141.00 | 141.22 | 141.21 | 141.09 | 141.13 | 0.16 |

| M2 | N2 | G2 | H2 | O2 | P2 | E2 | F2 | Average wall thickness | Wall thickness unevenness |
|---|---|---|---|---|---|---|---|---|---|
| 13.08 | 13.40 | 12.94 | 13.21 | 12.75 | 12.92 | 12.96 | 13.31 | 13.06 | 4.97 |

| Measurement section | | | | Average outside diameter | Ellipticity |
|---|---|---|---|---|---|
| M3-N3 | G3-H3 | O3-P3 | E3-F3 | | |
| 141.10 | 141.56 | 141.14 | 141.33 | 141.28 | 0.33 |

| M3 | N3 | G3 | H3 | O3 | P3 | E3 | F3 | Average wall thickness | Wall thickness unevenness |
|---|---|---|---|---|---|---|---|---|---|
| 13.00 | 13.34 | 12.71 | 13.22 | 12.60 | 12.86 | 13.15 | 13.03 | 12.99 | 5.71 |

| Measurement section | | | | Average outside diameter | Ellipticity |
|---|---|---|---|---|---|
| M4-N4 | G4-H4 | O4-P4 | E4-F4 | | |
| 141.01 | 141.16 | 141.19 | 141.01 | 141.09 | 0.13 |

| M4 | N4 | G4 | H4 | O4 | P4 | E4 | F4 | Average wall thickness | Wall thickness unevenness |
|---|---|---|---|---|---|---|---|---|---|
| 13.10 | 13.19 | 13.02 | 12.71 | 12.94 | 12.59 | 13.32 | 12.96 | 12.98 | 5.63 |

| Measurement section | | | | Average outside diameter | Ellipticity |
|---|---|---|---|---|---|
| M5-N5 | G5-H5 | O5-P5 | E5-F5 | | |
| 141.17 | 141.21 | 141.09 | 141.09 | 141.14 | 0.09 |

| M5 | N5 | G5 | H5 | O5 | P5 | E5 | F5 | Average wall thickness | Wall thickness unevenness |
|---|---|---|---|---|---|---|---|---|---|
| 13.33 | 13.20 | 12.96 | 12.96 | 13.15 | 12.65 | 13.10 | 13.00 | 13.02 | 5.23 |

Note: (1) Ellipticity calculation formula: $\frac{2(D_{max}-D_{min})}{D_{max}+D_{min}} \times 100\%$. Where: $D_{max}$—the maximum measured outer diameter value on the same cross-section; $D_{min}$—the minimum measured outer diameter value on the same cross-section. (2) Wall thickness unevenness calculation formula: $\frac{2(t_{max}-t_{min})}{t_{max}+t_{min}} \times 100\%$. Where: $t_{max}$—the same section on the measured maximum wall thickness value; $t_{min}$—the same section on the measured minimum wall thickness value.

Each collapsed specimen shall be subjected to residual stress measurement using the stress ring method, and the residual stress specimen shall be taken from the adjacent part of the collapsed specimen. The minimum length of the specimen should be two times the outer diameter (L/D ≥ 2), the specimen is shown in Figure 2, and the residual stress measurement results for specimen #1 are shown in Table 2.

**Table 2.** Residual stress measurement results.

| Specimen Number | Location of Measurements | Outer Diameter (mm) | | Wall Thickness (mm) | Residual Stress (MPa) |
|---|---|---|---|---|---|
| | | **B-D (before)** | **B-D (after)** | **C** | **/** |
| | | Di (m) | Df (mm) | t (mm) | / |
| | 1 | 141.29 | 142.16 | 13.04 | / |
| | 2 | 140.90 | 141.65 | 13.40 | / |
| 1# | 3 | 140.92 | 141.85 | 13.15 | / |
| | 4 | 140.90 | 141.78 | 13.06 | / |
| | Average value | 141.00 | 141.86 | 13.16 | 130.57 |

Note: The residual stress calculation formula: $\sigma = \frac{E_t}{(1-\mu^2)}\left(\frac{1}{D_i} - \frac{1}{D_f}\right)$. where $E = 2.1 \times 10^5$ MPa; $u = 0.3$.

The full-scale collapse test was conducted by an external pressure collapse test system shown in Figure 4. The full-scale collapse test specimens are shown in Figure 5. The collapse failure specimens are shown in Figure 6.

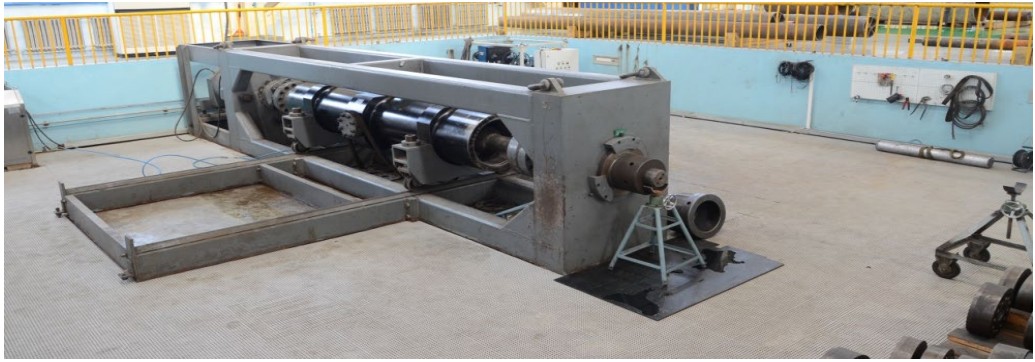

**Figure 4.** External pressure collapse test system.

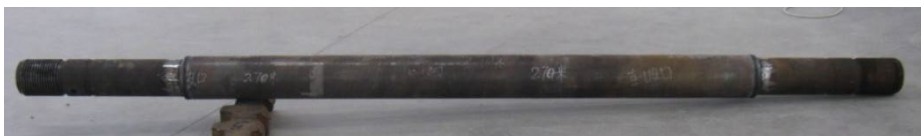

**Figure 5.** The specimens before collapse.

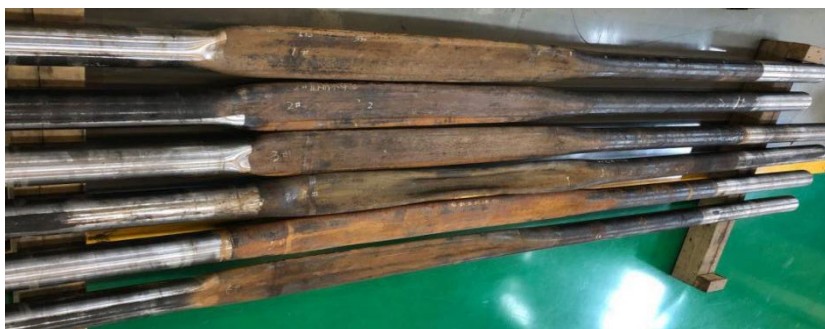

**Figure 6.** The specimens after collapse.

## 5. Establishment of Bayesian Regularized Neural Network for Prediction Casing Collapse Strength

### 5.1. Sample Data Pre-Processing

After the external extrusion test, the collected experimental data were grouped. Table 3 below shows the extrusion strength parameters corresponding to each pipe diameter of 4.5in, 5.5in, 7.0in, 9.5in, 13.5in, and 16in obtained after the experiment. Before model training, the data of each parameter should be unified in dimension to eliminate the useless data therein and form a casing collapse strength data set [24,25].

**Table 3.** Experimental data of casing collapse strength (partial).

| Outer Diameter in | Outer Diameter Out-of-Roundness % | Unevenness of Wall Thickness % | Residual Stress Mpa | Yield Strength Mpa | Casing Collapse Strength/psi |
|---|---|---|---|---|---|
| 4.53 | 0.471 | 3.145 | 14.76 | 700.53 | 12,469 |
| 4.51 | 0.416 | 2.356 | 116.92 | 817.06 | 13,550 |
| 4.51 | 0.074 | 4.400 | 225.95 | 661.92 | 13,416 |
| . . . | . . . | . . . | . . . | . . . | . . . |
| 5.53 | 0.37 | 0.643 | 156.86 | 464.03 | 4263 |
| 5.53 | 0.428 | 0.7 | 145.27 | 458.17 | 4160 |
| 5.53 | 0.405 | 0.376 | 153.87 | 468.86 | 4048 |
| . . . | . . . | . . . | . . . | . . . | . . . |
| 7.04 | 0.194 | 2.872 | 62.85 | 498.85 | 5757 |
| 7.05 | 0.192 | 2.304 | 130.21 | 480.92 | 5961 |
| 7.05 | 0.165 | 6.662 | 108.27 | 490.94 | 5400 |
| . . . | . . . | . . . | . . . | . . . | . . . |
| 9.70 | 0.359 | 3.678 | 105.48 | 452.66 | 2689 |
| 9.71 | 0.402 | 1.985 | 35.31 | 645.37 | 5032 |
| 9.71 | 0.401 | 2.781 | 34.83 | 649.509 | 4975 |
| . . . | . . . | . . . | . . . | . . . | . . . |
| 13.47 | 0.368 | 1.987 | 213.31 | 900.00 | 3342 |
| 13.47 | 0.184 | 1.643 | 254.84 | 890.48 | 3316 |
| 13.44 | 0.212 | 1.603 | 172.45 | 886.35 | 3208 |
| . . . | . . . | . . . | . . . | . . . | . . . |
| 16.09 | 0.231 | 2.329 | 31.351 | 721.56 | 2669 |
| 16.08 | 0.291 | 4.723 | 37.79 | 718.45 | 2550 |
| 16.08 | 0.234 | 4.682 | 29.73 | 712.59 | 2413 |

In order to increase the validity of data, here, the min-max (Min-Max Normalization) standardized data consistency processing method will be selected. This method, also called deviation standardization, is a linear transformation of the original data, so that the result value is mapped to [0–1]. The conversion function is as follows:

$$X^* = \frac{X - min}{max - min} \tag{13}$$

where *max* is the maximum value of the sample data and *min* is the minimum value of the sample data. The drawback of this method is that once new data are added, it may lead to changes in *max* and *min* and therefore needs to be redefined.

### 5.2. Define the Network Structure

The determination of the number of layers of the neural network model, i.e., the determination of the hidden layers. In this paper, a three-layer neural network structure (one input layer, one hidden layer, and one output layer) will be selected. The number of nodes in the input and output layers is equal to the number of input and output parameters, and the optimal number of neurons in the hidden layer is obtained by comparing them after several training sessions. The five main correlation parameters of casing collapse strength, namely diameter-thickness ratio, ellipticity, wall thickness inhomogeneity, yield strength, and residual stress, will be used as the input of the neural network, and the output will be the casing collapse strength. Under such a three-layer neural network, the input layer contains five neurons, and the output layer contains one neuron. The structure of this neural network is shown in Figure 7 below.

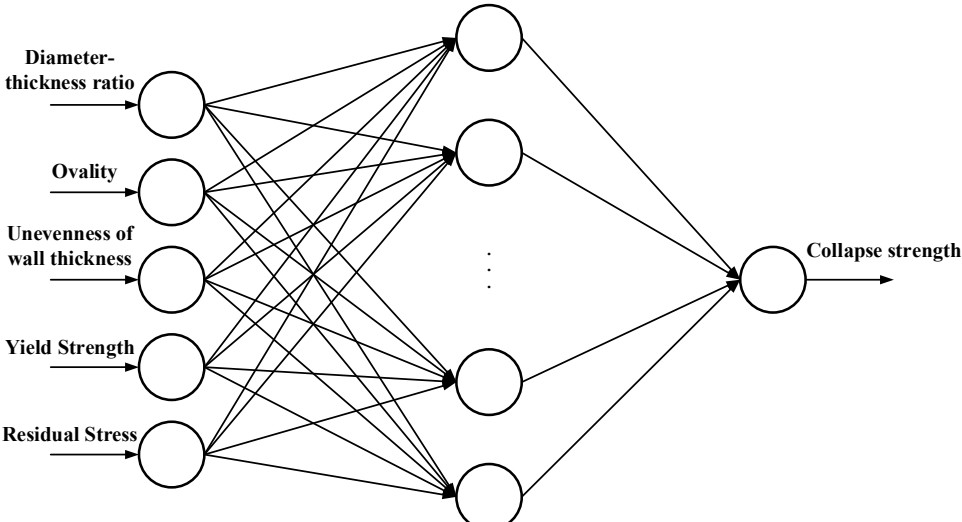

**Figure 7.** Neural network structure for prediction the casing collapse strength.

### 5.3. Model Training and Optimization

Among the obtained data, 2/3 of the data are selected for model development, and 1/3 (not involved in model training) are used to test the degree of generalization of the developed model, and the algorithm is selected as a Bayesian regularization algorithm. In the model development group data, 85% of the data will be randomly divided into training set data, and 15% of the data will be test set data, where each group of data includes 5 inputs (diameter-thickness ratio, ellipticity, wall thickness unevenness, yield strength, and residual stress) and one output (casing collapse strength).

In the optimization of the model, the mean square error (MSE) and the decision coefficient ($R^2$) were mainly referenced. The MSE was the average square deviation between the output value and the target value. The value of $R^2$ was used to measure the correlation between the output value and the target value. When the $R^2$ of the training set was greater than that of the test set, it indicated that an over-fitting occurred. When the model with the MSE as low as possible and the $R^2$ value close to 1 is the appropriate model in the testing stage, these two indicators can show whether the model extracts all the information or whether further adjustment is required. The fitting effect of the model is poor when the number of neurons is too small, but it will lead to over-fitting. In order to find the optimal model, the purpose can be achieved by adjusting the sample size, the proportion of the dataset, or the number of neurons in the hidden layer. In order to obtain effective feedback and eliminate the data deviation caused by network training fluctuation, the training process would undergo 10 training and blind tests under the same condition and the average value would be taken. In view of the input parameter $n = 5$ of the model, the selection of the number of neurons in the hidden layer N follows $N \geq n$, and the trial-and-error initial value of the number of neurons in the hidden layer $N_0 = 5$, the trial-and-error upper limit value is set according to the empirical formula method (Equation (14)).

$$N = \sqrt{n + m} + a \tag{14}$$

where n is the number of nodes in the input layer, m is the number of nodes in the output layer, and a is an integer from 1 to 10.

Table 4 shows the average $R^2$ of the neural network model after 10 times of training under different numbers of hidden neurons, and Figure 8 shows the comparison curve of the determination coefficient $R^2$ of model training and prediction when the number of hidden neurons is 5~15.

**Table 4.** $R^2$ of the model with different numbers of hidden layer neurons.

| Number of Neurons in the Hidden Layer | Neural Network Model Training $R^2$-Value | Neural Network Model Predicting $R^2$-Value |
|---|---|---|
| 5 | 0.99699 | 0.99715 |
| 6 | 0.99707 | 0.99730 |
| 7 | 0.99675 | 0.99727 |
| 8 | 0.99689 | 0.99701 |
| 9 | 0.99537 | 0.99774 |
| 10 | 0.99716 | 0.99748 |
| 11 | 0.99684 | 0.99750 |
| 12 | 0.99746 | 0.99780 |
| 13 | 0.99740 | 0.99742 |
| 14 | 0.99712 | 0.99775 |
| 15 | 0.99685 | 0.99754 |

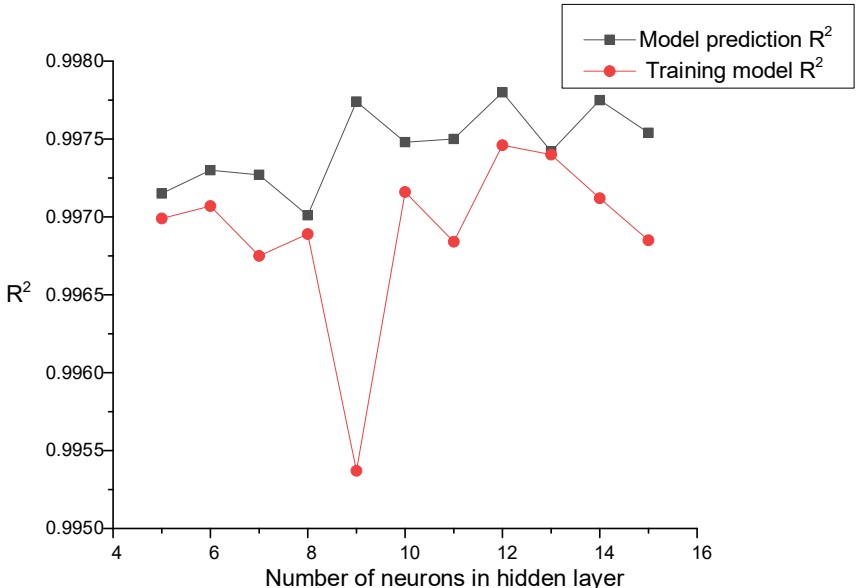

**Figure 8.** $R^2$ comparison of the model with 5–15 hidden neurons.

It can be seen from Table 4 and Figure 8 that the change in the number of neurons in the hidden layer affects the prediction accuracy of the model when the proportion of data sets is constant. There is no regular relationship between the $R^2$ value and the number of neurons in the hidden layer, but overall, the determining coefficient $R^2$ tends to approach 1. It can be clearly seen from the comparison curve in Figure 8 that the predicted $R^2$ of 11 groups of models is higher than that of model training $R^2$, and the model has not been fitted. When the number of hidden neurons is 9, the difference between the two $R^2$ values is the biggest, and when the number of hidden neurons is 13, $R^2$ is the closest. When the number of hidden layer neurons is 12, the average value of the training $R^2$ and the prediction $R^2$ of the model is closest to 1. In addition, as can be seen from Figure 9, the error of the network model is the smallest compared with other models at this time, which best meets the demand of model optimization. Therefore, the number of hidden layer neurons N is set to 12 for prediction research.

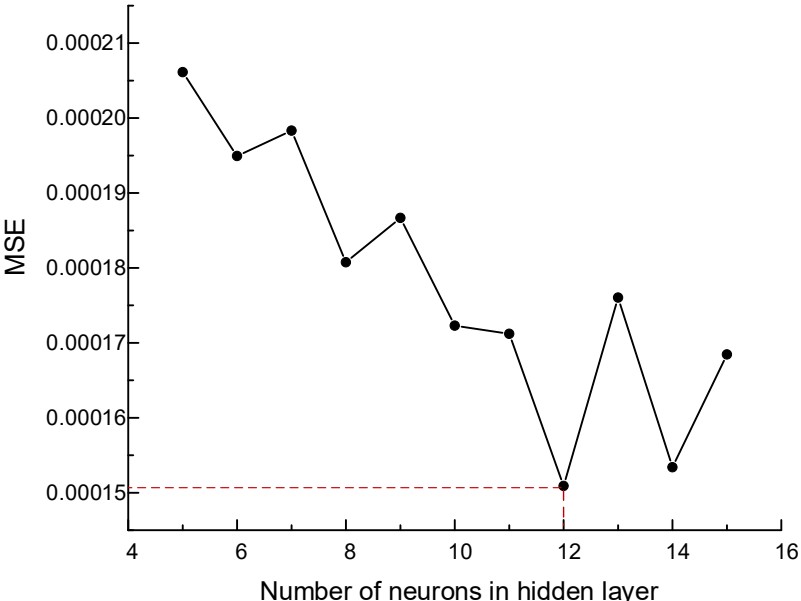

**Figure 9.** Network training error when the number of hidden layer neurons is between 5–15.

### 5.4. Model Evaluation and Prediction Result Analysis

According to 5.3, the number of hidden layer neurons N = 12 is set to predict the casing collapse strength. To test the effectiveness and advantages of the model, the casing collapse pressure is calculated according to API 5C3 specification on the basis of the measured data, and combined with KT improved formula in ISO/TR 10400:2007 specification, the minimum collapse pressure of casing is predicted by the least square regression method. For the casing with D/T < 12.53, the collapse strength is related to the collapse pressure Py of the yield strength, which is calculated by Equation (15).

$$P_y = 2\sigma_s \left[ \frac{(D/t) - 1}{(D/t)^2} \right] \tag{15}$$

When 12.53 < D/t < 20.56, the minimum collapse strength is related to the plastic collapse pressure *Py*, which is calculated by Equation (16).

$$P_p = \sigma_s \left[ \frac{A}{D/t} - B \right] - C \tag{16}$$

$$A = 2.8762 + 0.15489 \times 10^{-3}\sigma_s + 0.44809 \times 10^{-6}\sigma_s^2 - 0.16211 \times 10^{-19}\sigma_s^3 \tag{17}$$

$$B = 0.026233 + 0.73402 \times 10^{-4}\sigma_s \tag{18}$$

$$C = -3.2125 + 0.030867\sigma_s - 0.15204 \times 10^{-5}\sigma_s^2 + 0.7781 \times 10^{-9}\sigma_s^3 \tag{19}$$

When D/t > 20.56, the minimum collapse strength is related to the elastic collapse pressure *Py*, which is calculated by the elastic collapse pressure Equation (20).

$$P_E = \frac{3.237 \times 10^5}{(D/t)[(D/t) - 1]^2} \tag{20}$$

The data set to be measured includes 4.5in, 5.5in, 7.0in, 9.5in, 13.5in, 16in, and other pipe diameter parameters. According to the BRANN model, the data to be predicted is imported for prediction, and regression fitting and formula calculation are carried out on the data to be measured. Figure 10 below shows the comparison between the model prediction results, regression prediction results, formula calculation values, and measured collapse strength, and Figure 11 shows the error distribution curves of the results obtained

by three methods, in which the error is calculated by the relative error method, that is |
predicted value-actual value |/100% of actual value.

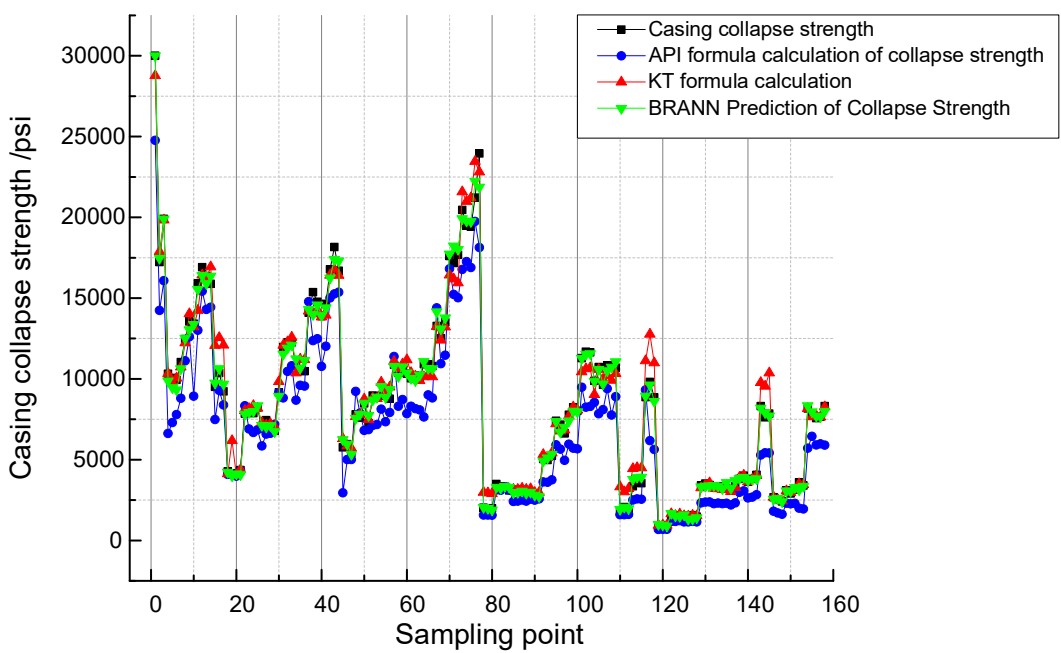

**Figure 10.** Prediction result of casing collapse strength.

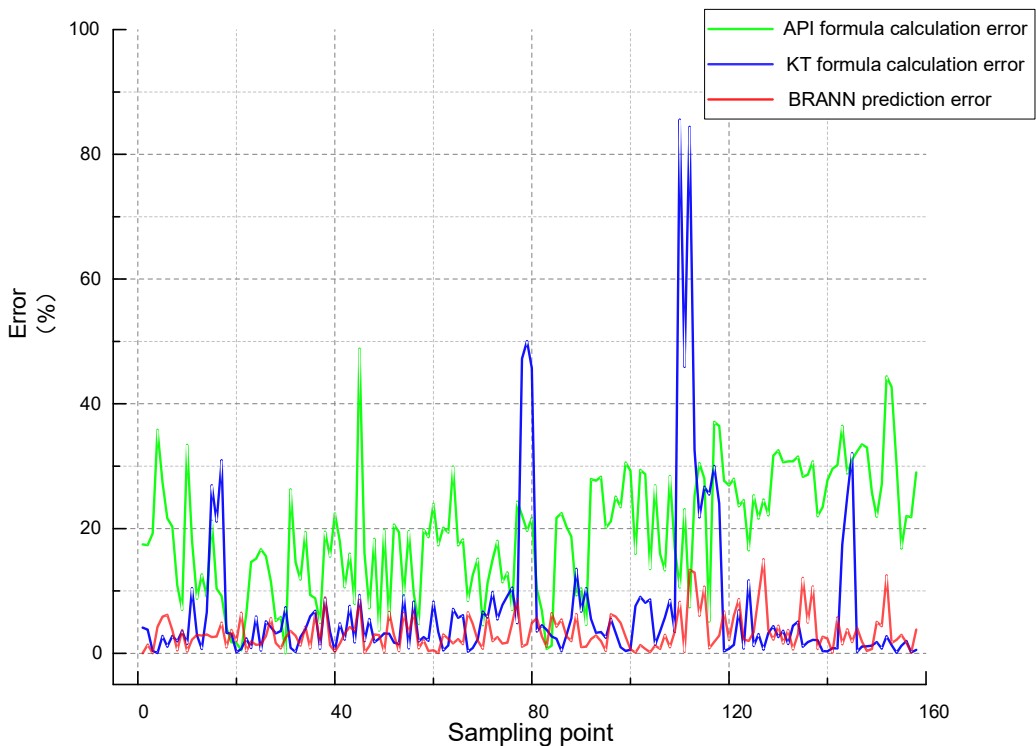

**Figure 11.** Comparison of prediction errors of collapse strength.

From the distribution of prediction results, it can be seen that under different pipe
diameters, the calculated results guided by API specifications are obviously deviated from
the measured values. Both the least square regression fitting and Bayesian neural network
can predict the casing collapse strength. Compared with the traditional formula calculation,
the coincidence rate between the two prediction results and the measured values is higher.

As for the trend of errors, the following Table 5 shows the maximum, minimum, and average values of each error.

**Table 5.** Output error of blind test samples.

| Tape | Max | Min | Average |
|---|---|---|---|
| API formula | 48.84% | 0.02% | 19.46% |
| The least square improved KT formula | 85.56% | 0.06% | 7.41% |
| BRANN | 15.09% | 0.01% | 3.33% |

Combined with Figure 11 and Table 5, for the same blind sample input, the minimum errors of the results obtained by the three methods are all less than 0.1%. In terms of stability, the error span of the least square regression fitting results is the largest, with the maximum error reaching 85.56%. By comparing the sample information, it is found that the error is mainly distributed in the samples with a diameter of 10.8 inches. For the pipe fittings with a diameter of 5 inches to 7.8 inches, the least square fitting effect is relatively stable. It can be seen from Figure 10 that the error of API formula calculation results shows an obvious swing between 0.02% and 50%, and the average error reaches 19.46% in different pipe diameters; Combined with the error curve distribution in Figure 10, the error trend of BRANN model is more stable than that of the other two methods, with the maximum error of 15.09%. In different pipe diameters, the model can achieve good prediction results, and the average prediction accuracy of the model can reach 96.67%, except that the prediction error of individual samples in the range of 10.8~14.4in inches is more than 10%, which is significantly improved compared with the traditional methods.

## 6. Conclusions

(1) The experimental results of five parameters affecting the casing collapse strength (diameter-thickness ratio, ellipticity, wall thickness unevenness, yield strength, and residual stress), are obtained by full-scale tests, which constitute the training data set of BRANN.

(2) The number of hidden layers neurons is 12, the R-value is closest to 1, which meets the prediction accuracy requirement.

(3) Based on the established BRANN model, the casing collapse strength prediction and supplementary regression fitting were performed. The results show that the BRANN-based prediction model has higher prediction accuracy, and the maximum error between this model and the physical experimental test results is 13.11%, and most of the errors are less than 10%.

**Author Contributions:** Conceptualization, X.Y.; methodology, D.L. and H.F.; validation, S.Y.; formal analysis, Y.Z. and D.L.; data curation, R.W.; writing—original draft preparation, Y.Z.; writing—review and editing, H.F. All authors have read and agreed to the published version of the manuscript.

**Funding:** This research was funded by the Innovative Talents Promotion Program—Young Science-and Technology Nova Project (2021KJXX-63), the Research on key technology of casing damageevaluation and repair in oil and gas wells (2021DJ2705) and the Study on key technology of stimulationand modification for Gulong shale oil (2021ZZ10-04).

**Institutional Review Board Statement:** Not applicable.

**Informed Consent Statement:** No involving humans.

**Data Availability Statement:** The study did not report any data.

**Conflicts of Interest:** The authors declare no conflict of interest.

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
