# Peer review of "Prediction of Casing Collapse Strength Based on Bayesian Neural Network"

_processes, doi:10.3390/pr10071327_

Round 1

Reviewer 1 Report

I congratulate the authors on their work. They clearly know the subject and develop well the laboratory tests in conjunction with the BRANN-based model for external strength control. Once an accident occurs, a spill can escalate a major fire and explode, and pose a risk to human safety, the environment, property and reputation. The risk of a pipeline failure cannot be eliminated, but preventive and mitigation measures can be taken to reduce the likelihood and severity of the consequences of a leak. Risk analysis is an effective tool for identifying risk factors and developing accident prevention strategies. The authors of the study contributed to the analysis of the risk of pipeline failure, presented the basic method of risk analysis for pipelines that may cause their damage. In this method, the probabilities of various failure mechanisms are calculated using the limit state function, while the consequences of failure are measured with financial losses. The authors developed an experimental and numerical model for pipeline damage risk assessment, which is estimated by sampling the scenario, and the consequences of the accident are simulated using the finite element method. In addition, the authors presented an integrated evaluation method based on the BRANN model, external extrusion sheath strength as a control, prediction and complementary regression fit. Also, studies have shown that the BRANN-based predictive model has a higher maximum error between this model and the experimental test is 13.11%, and most errors are less than 10%, which meets the engineering requirements. The article contains minor grammatical errors that should be corrected. Requests that the article be accepted for publication.

Author Response

Thank you very much. The grammar has been modified

Reviewer 2 Report

·         Re-write introduction. There are important studies (related to your study) that were not cited in the manuscript. A comprehensive review of published literature on the study subject is required.

·         Compare your method (ANN) used in this study to other published studies in terms of its advantages and drawbacks.

·         Present a research flow chart for your study.

·         How did you optimize the number of neurons and hidden layers? Please show the algorithm.

·         I would recommend depicting your outputs versus input variables to see the effect of boundaries on the performance of the machine learning model

Author Response

Response to Reviewer:

Reviewer #: Re-write introduction. There are important studies (related to your study) that were not cited in the manuscript. A comprehensive review of published literature on the study subject is required.

Q1. Compare your method (ANN) used in this study to other published studies in terms of its advantages and drawbacks.

Answer: In combination with that traditional prediction method of casing collapse strength, API formula and KT formula are added to the samples to be predicted in the model evaluation part, and the model propose in this paper is evaluated by horizontally comparing the prediction errors of BRANN, API formula and KT formula.

Q2. Present a research flow chart for your study.

Answer: The flow chart of casing collapse strength prediction scheme (Figure 1) is supplemented in 2.1.

Q3. How did you optimize the number of neurons and hidden layers? Please show the algorithm.

Answer: The number of neurons in the input and output layers was set according to the selected casing collapse strength prediction parameters. The number of neurons in the hidden layer combined the empirical formula method and the trial-and-error method. For five input parameters and one output parameter, the initial value of the hidden layer neuron test was set to 5. The neurons were accumulated in sequence to train the model. The optimal number of hidden layer neurons was selected by comparing the MSE and R2 of the model. See 4.3 for details.

Q4. I would recommend depicting your outputs versus input variables to see the effect of boundaries on the performance of the machine learning model

Answer: In this study, the selection of input and output variables refers to the key influencing parameters of the object to be predicted. Because the model training needs a certain amount of data, the selection of the number of variables needs to be combined with the difficulty and validity of data acquisition. In this study, the input variables mainly extract the geometric size and manufacturing defect parameters of the casing itself, and more dimensional input variables cannot be reflected because of the difficulty in obtaining them. In the future, we will continue to study the prediction methods of casing strength under the influence of multiple factors.

See attachment for detailed modification,Thank you.

Round 2

Reviewer 2 Report

Accept in present form